# Thermal Characteristics of Spindle System Based on the Comprehensive Effect of Multiple Nonlinear Time-Varying Factors

**Xiaoliang Lin [1], Xiaolei Deng [1],\*, Junjian Zheng [2], Xinhua Yao [3] and Hongyao Shen [3]**

[1] Key Laboratory of Air-Driven Equipment Technology of Zhejiang Province, Quzhou University, Quzhou 324000, China; lxl@qzc.edu.cn
[2] College of Mechanical Engineering, Zhejiang University of Technology, Hangzhou 310023, China; 221122020255@zjut.edu.cn
[3] Key Laboratory of 3D Printing Process and Equipment of Zhejiang Province, State Key Laboratory of Fluid Power and Mechatronic Systems, School of Mechanical Engineering, Zhejiang University, Hangzhou 310027, China
\* Correspondence: dxl@zju.edu.cn

**Abstract:** The thermal characteristics of the spindle system for CNC machine tools are influenced by multiple factors which are nonlinear and time-varying. In this paper, a nonlinear time-varying thermal characteristics solving model for the spindle system was established based on the numerical solution method. Through theoretical deduction and data fitting, mathematical models of nonlinear time-varying factors including the friction torque generated by lubricants, convective heat transfer coefficient, and coolant and ambient temperature are constructed. The temperature and displacement of the spindle system at each time step are solved by considering the comprehensive effect of multiple nonlinear time-varying factors. And the actual temperature and axial deformation data of the spindle system are obtained through thermal characteristics detection experiments. By comparing solution results affected by multiple nonlinear time-varying factors and by non time-varying factors with experimental data, it can be concluded that the nonlinear time-varying thermal characteristics model has advantages in reflecting the trend of numerical changes and the accuracy of result solving over a method considering non time-varying factors and the solution values of temperature affected by multiple nonlinear time-varying factors are almost consistent with detection values and the relative errors are all within ±3%. The relative error of axial deformation between the value solved by the model and the detection value is close to −1%. This conclusion demonstrates the rationality and accuracy of the thermal characteristics solving model and the construction of nonlinear time-varying factors. This study is of great significance for exploring the thermal characteristics of the spindle system and improving CNC machine tool performance in depth.

**Keywords:** CNC machine tool; spindle system; thermal characteristics; nonlinear time-varying factors; multiple factors; comprehensive effect

## 1. Introduction

The technical level of CNC machine tools directly affects the development of the manufacturing industry. The accuracy and stability of CNC machine tools are crucial for product quality [1]. The spindle system, whose accuracy is one of the core indexes to measure the performance of a CNC machine tool, is the end effector during the machining process of the workpiece [2]. As a complex assembly composed of spindle, bearing supports, lubrication, cooling, and other component structures, its machining error is the result of the comprehensive effect from multiple factors related to the components and their interrelationships [3]. Heat-related factors account for 40% to 70% of all factors which cause accuracy loss of the spindle system [4,5]. Starting from the factors affecting the thermal

characteristics to study their comprehensive effect on the temperature and deformation of the spindle system is an effective method which can improve the performance of CNC machine tools.

In recent years, there have been numerous studies on the thermal characteristics and their influencing factors on the spindle system. Lee et al. [6] researched the changes in temperature and deformation of the spindle system due to the influence of spindle speed by the finite element method and thermal analysis method. The relationship between thermal deformation and vibration was analyzed and modeled, which was verified experimentally. Raja et al. [7] developed a coupled fluid–thermal model for spindles. The model can simulate the fluid–structural conjugate heat transfer of the spindle system. They measured the minimum deviation of the model through experiments as 7.6%. Marez et al. [8] developed an approach based on TFs which is used for effective thermal error modeling for machine tools. This approach provides insight into the share of each source in the total machine thermal error through a combination of linear parametric models. Kaftan and Wegener et al. [9] studied the adverse effects of internal and external heat sources on the complex non-symmetric structure of machine tools. They developed a novel method based on artistic intelligence that compensates for thermal errors associated with hidden boundary condition changes to address the drawbacks of traditional methods. Tan et al. [10] studied the thermal characteristics of spindle bearings under preloading and achieved nonlinear prediction of thermal characteristics. The nonlinear changes in contact angle, contact force, preload, and stiffness were analyzed by considering the effects of contact thermal resistance, bearing parameters, lubricant viscosity, and time-varying temperature on the heat source. The proposed method significantly reduced the computational workload. Liu and Ma et al. [11] proposed a closed-loop iterative modeling method. The heat generation of bearings and built-in motors, convection coefficient of bearing joints, and thermal contact resistance were calculated by modifying the heat source and thermal boundary conditions in each calculation step. By considering the comprehensive effects of lubricant viscosity changes and bearing thermal preload, the bearing heat generation was corrected. Xiang et al. [12] proposed a data-driven prediction approach which can establish a dynamic linear model of spindle thermal error. The serious contradictions in traditional prediction methods have been resolved by predicting current thermal errors through historical temperature data without physical mechanism information. Wu et al. [13] established a thermo-mechanical coupling analysis model for the spindle bearing system of machine tools. Through this model, they obtained the relationship between bearing preload and frictional heat generation, as well as between coolant and system thermal balance. Brecher, Wenkler, and Ihlenfeldt et al. [14,15] also explored the thermal related influencing factors of CNC machine tools in the research of thermal error modeling.

After the CNC machine tool is started, its state and surrounding environment exhibit nonlinear time-varying characteristics. The comprehensive effect of multiple changing factors is an important cause for the poor thermal characteristics and accuracy loss of CNC machine tools. At present, according to the research of many scholars, the research on factors such as heat sources, bearing loads, and coolant is relatively in-depth. The nonlinear and time-varying effects of these factors on the spindle have also been studied separately. However, there are few reports on solving the thermal characteristics of the spindle system under the comprehensive effects of multiple nonlinear time-varying factors such as heat sources, cooling conditions, surrounding environment, and heat transfer capability. Therefore, this paper establishes a numerical solving model for the nonlinear time-varying thermal characteristics of the spindle system. Based on the time-varying variables of the model, mathematical models of nonlinear time-varying factors such as friction torque generated by lubricant, convective heat transfer coefficient, and coolant and ambient temperature are constructed through theoretical derivation and data fitting. The time-varying temperature of the spindle system is solved taking into account multiple factors. Based on the time-varying value of the temperature, the deformation of the spindle

at each moment is calculated. This paper provides a theoretical basis for the thermal characteristics study of spindle systems under complex working conditions.

## 2. Establishment of the Nonlinear Time-Varying Thermal Characteristics Model

During the operation of CNC machine tools, the heat transfer process of mechanical components in the spindle system follows the control equation shown in Equation (1) [16].

$$\rho C_P \frac{\partial T}{\partial t} = \nabla \cdot (k \nabla T) + S \tag{1}$$

where $\rho$ is the material density of the component, $C_P$ is specific heat, $T$ is temperature, $t$ is time, $k$ is thermal conductivity, and $S$ is source terms.

Based on the finite volume method, this study will use hybrid grids to partition the control volume of the physical model for the spindle system. For each control volume, the central difference scheme and time implicit format are introduced. The pressure-based solver of Fluent 2022 R2 software will be used for solving. The least square cell-based and second-order upwind are adopted for spatial discretization. Therefore, within the allowable range of discretization error and numerically controllable, Equation (1) can be expanded as

$$\rho C_P (T_P^{t_i} - T_P^{t_{i-1}}) V_P = \sum_{N=1}^{M} k_{PN} \frac{T_N^{t_i} - T_P^{t_i}}{\delta L_{PN}} A_{PN} \Delta t_i + s V_P \Delta t_i \tag{2}$$

where $T_P^{t_i}$ and $T_P^{t_{i-1}}$ are the average temperature of the local control volume $P$ at the $i$-th and $i-1$th time step, $V_P$ is the volume of $P$, $M$ is the number of adjacent control volumes to $P$, $T_N^{t_i}$ is the average temperature of control volume $N$ adjacent to $P$ at the $i$-th time step, $k_{PN}$ is the harmonic mean of thermal conductivity between control volume $P$ and $N$, $\delta L_{PN}$ is the distance between the center of $P$ and $N$, $A_{PN}$ is the heat transfer surface area between $P$ and $N$, $s$ is the intensity of the heat source, $\Delta t_i$ is the $i$-th time step size, and $\Delta t_i = t_i - t_{i-1}$.

The heat flux on both sides of the interface between the spindle system and the external fluid zone in contact with the system is equal under the state of fluid–solid coupling heat transfer [17]. According to Fourier's law and Newton's flow equation, Equation (3) can be obtained.

$$-k \frac{\partial T}{\partial L} = h \Delta T \tag{3}$$

where $L$ is the distance along the vector normal to the interface, $h$ is the convective heat transfer coefficient, and $\Delta T$ is the temperature difference between the fluid and wall surface.

Assuming the temperature of fluid in contact with the spindle system is $T_F$, Equation (4) can be established within the allowable range of discretization error and numerically controllable.

$$-k_P \frac{T_W^{t_i} - T_P^{t_i}}{\delta L_{PW}} = \frac{T_F - T_W^{t_i}}{1/h_F} \tag{4}$$

where $k_p$ is the thermal conductivity of $P$, $T_W^{t_i}$ is the temperature of the wall surface at the $i$-th time step, $\delta L_{PW}$ is the distance between the center of $P$ and the wall surface, and $h_F$ is the convective heat transfer coefficient of the fluid.

In components, there is obviously Equation (5), as follows.

$$k_{PN} \frac{T_N^{t_i} - T_P^{t_i}}{\delta L_{PN}} = k_P \frac{T_W^{t_i} - T_P^{t_i}}{\delta L_{PW}} \tag{5}$$

Take $\delta L_{PN} = 2\delta L_{PW}$. According to Equations (4) and (5) above, Equation (6) can be obtained.

$$-k_{PN} \frac{T_N^{t_i} - T_P^{t_i}}{\delta L_{PN}} = \frac{T_F - T_P^{t_i}}{(1/h_F) + (2\delta L_{PN}/k_P)} \tag{6}$$

It is assumed that the thermal conductivity of the spindle system material is constant. The nonlinear time-varying thermal characteristics of spindle systems should be considered from heat generation and heat dissipation. The heat generation is reflect in the intensity of the heat sources whose values at the $i$-th time step can be expressed as $s^{t_i}$ in the control equation. And the heat dissipation is usually related to the external ambient conditions and the capabilities to transfer heat outward, which is mainly reflected in the convective heat transfer coefficient and the temperature of the fluid in contact with the system. Their values at the $i$-th time step can be expressed as $h_F^{t_i}$ and $T_F^{t_i}$ in the control equation. By combining Equations (2) and (6), the nonlinear time-varying thermal characteristics model shown in Equation (7) can be derived through substituting $s^{t_i}$, $h_F^{t_i}$, and $T_F^{t_i}$.

$$\rho C_p(T_P^{t_i} - T_P^{t_{i-1}})V_P = \sum_{N=1}^{M-J} k_{PN}\frac{T_N^{t_i} - T_P^{t_i}}{\delta L_{PN}}A_{PN}\Delta t_i + \sum_{F=1}^{J}\frac{T_F^{t_i} - T_P^{t_i}}{[1/h_F^{t_i}] + (2\delta L_{PN}/k_P)}A_{PF}\Delta t_i + s^{t_i}V_P\Delta t_i \tag{7}$$

where $J$ is the number of convective heat transfer surfaces of $P$, and $A_{PF}$ is the convective heat transfer surface area of control volume $P$.

The heat transfer equation shown in Equation (7) is the model for the nonlinear time-varying thermal characteristics of the spindle system. According to the model, $s^{t_i}$, $h_F^{t_i}$, and $T_F^{t_i}$ are the variables which can affect the value of the temperature of control volume $P$ and its adjacent control volume $N$. They are usually nonlinear and time-varying. As the definite solution conditions for the temperature of the spindle system, the models of nonlinear and time-varying factors related to heat source intensity, convective heat transfer coefficient, and the temperature of fluid in contact with the system should be constructed and substituted into the solution.

### 3. Construction of Models for Nonlinear Time-Varying Factors inside and outside the System

*3.1. Time-Varying Factors of System Heat Source*

For non-heating component in the spindle system, $s(t) = 0$ and for components that generate heat, $s(t)$ is obtained according to Equation (8).

$$s(t) = \frac{Q(t)}{V} \tag{8}$$

where $Q(t)$ is the time-varying function of heat flow from the heating component, and $V$ is the volume of the heat source component.

The heating of the motor and bearings is the main cause of temperature rise in the spindle system [18,19]. Due to the installation of the motor outside the spindle box, less heat is transferred to the spindle components through the mounting surface of the box. The bearings heating is due to the viscosity of lubricants and the torque generated by bearing loads. The heat flow $Q$ can be expressed by Equation (9) [11].

$$Q = 1.047 \times 10^{-4}(M_0 + M_1)n \tag{9}$$

where $M_0$ is the friction torque generated by lubricant viscosity, $M_1$ is the friction torque generated by the load on the bearing, and $n$ is the spindle speed.

During the rotation of bearing, the viscosity of the lubricant changes with the increase in temperature. In other words, if oil lubrication is used, the viscosity $v_{\text{oil}}$ is a time-varying factor related to temperature $T_{\text{oil}}$, as shown in Equation (10).

$$v_{\text{oil}} = v_{\text{oil}}(t) = v_{\text{oil}}[T_{\text{oil}}(t)] \tag{10}$$

The spindle rotates at a speed of $n$. $M_0$ generated by lubricant viscosity exhibits time-varying and nonlinear characteristics. When $nv_{oil} \geq 2000$, Equation (11) can be obtained according to references [20].

$$M_0(t) = 10^{-7}f_0\{v_{oil}[T_{oil}(t)]n\}^{2/3}d_m^3 \tag{11}$$

where $M_0(t)$ is the time-varying function of the friction torque generated by lubricant viscosity, $f_0$ is the coefficient determined by bearing structure and lubrication, and $d_m$ is the middle diameter of bearing.

From Equations (9) and (11), it can be seen that the heat generated by the lubricant will change the viscosity of the lubricant, whose change will also affect the heat generation in turn. The two mutually constrain each other through the intermediate variable $T_{oil}$ $(t)$. The value of $T_{oil}$ $(t)$ is obtained by replacing $T_F^{t_i}$ and iteratively calculating based on the nonlinear time-varying thermal characteristics model shown in Equation (7). The coupling relationship of $v_{oil}$, $T_{oil}$, and $M_0$ makes them exhibit time-varying and nonlinear characteristics.

### 3.2. Time-Varying Factors of Heat Dissipation outside the System

The heat dissipation outside the spindle system mainly involves convection heat transfer with air forcefully and naturally. The air characteristics affected by ambient temperature are nonlinear and time-varying due to the long time from startup to thermal equilibrium of the spindle system, which causes both the convective heat transfer coefficient and the temperature difference inside and outside the system to change simultaneously. In other words, the construction of time-varying factor models for the external heat dissipation of the system should include ambient temperature and convective heat transfer coefficient.

Forced convection heat transfer mainly occurs on the rotating surface of the spindle system. When the Reynolds number $Re_{air}$ and the Prandtl number $Pr_{air}$ satisfy $Re_{air} < 4.3 \times 10^5$ and $0.7 < Pr_{air} < 670$, according to reference [21], the forced convection heat transfer coefficient $h_f(t)$ can be calculated using Equation (12).

$$h_f(t) = \frac{Nu_{air}(t)\lambda_{air}(t)}{d_s} \tag{12}$$

where $Nu_{air}(t)$ is the Nusselt number that varies over time, $\lambda_{air}(t)$ is the time-varying function of air thermal conductivity, and $d_s$ is the equivalent diameter of rotating surfaces.

According to references [22,23], the forced convection heat transfer coefficient $h_f(t)$ can be derived by expressions of the Nusselt number as follows:

$$h_f(t) = 0.133\left[\frac{\pi\rho_{air}(t)n}{60\mu_{air}(t)}\right]^{2/3}[d_s Pr_{air}(t)]^{1/3}\lambda_{air}(t) \tag{13}$$

where $\rho_{air}(t)$, $\mu_{air}(t)$, and $Pr_{air}(t)$ are the time-varying functions of density, kinematic viscosity, and the Prandtl number for air, respectively.

For spindle segments with different diameters, the equivalent diameter $d_s$ can be calculated by Equation (14) [24]:

$$d_s = \frac{1}{l}\sum_{i=1}^{n}d_i l_i \tag{14}$$

where $l$ is the total length of spindle, and $d_i$ and $l_i$ are the diameter and the length of the $i$-th spindle segment.

For natural convection heat transfer surfaces, the time-varying function of the natural convection heat transfer coefficient $h_n(t)$ can be derived according to Equation (15) [20].

$$h_n(t) = 0.59\left[\frac{g\beta_{air}(t)Pr_{air}(t)\Delta T(t)}{D}\right]^{1/4}\frac{\lambda_{air}(t)}{\mu_{air}(t)^{1/2}} \tag{15}$$

where $g$ is the standard gravitational acceleration, $\beta_{\text{air}}(t)$ is the time-varying function of air thermal expansion coefficient, $D$ is the feature size, and $\Delta T(t)$ is the temperature difference between ambient air and the heat transfer wall, which is obtained using the following equation:

$$\Delta T(t) = T_W(t) - T_{\text{air}}(t) \tag{16}$$

where $T_{\text{air}}(t)$ is the time-varying function of ambient temperature, and $T_W(t)$ is the time-varying function of the heat transfer wall surface and its value determined by Equation (5).

### 3.3. Time-Varying Factors of Internal Cooling inside the System

The convective heat transfer of the spindle system not only occurs externally with the air but also internally with the coolant such as cooling oil. Similar to the time-varying factors of external heat dissipation, the time-varying characteristics of internal factors such as convective heat transfer coefficient $h_{\text{oil}}$ and cooling oil temperature $T_{\text{oil}}$ need to be considered.

Due to the spiral-shaped slender channel in the cooling jacket of spindle system, the flow of cooling oil is mostly laminar. This paper assumes that the oil density remains constant. The nonlinear time-varying function of the convective heat transfer coefficient $h_{\text{oil}}$ of cooling oil can be constructed as Equation (17) based on reference [25].

$$h_{\text{oil}}(t) = 1.86 \left( \frac{C_{P\text{oil}}(t) \rho_{\text{oil}} d_c^2}{\lambda_{\text{oil}}(t) l_c} \right)^{1/3} \tag{17}$$

where $C_{P\text{oil}}(t)$ is the time-varying function of oil-specific heat, $\rho_{\text{oil}}$ is the density of oil, $d_c$ is the hydraulic diameter of cooling channel, and $\lambda_{\text{oil}}(t)$ is the time-varying function of oil thermal conductivity. $l_c$ is the channel length.

The nonlinear time-varying function $T_{\text{oil}}(t)$ of the cooling oil temperature is obtained by fitting the data detected by the experiment.

## 4. Experimental Platform Construction and Data Detection

This paper conducts experimental research on the G1160 CNC machine tool. This CNC machine tool adopts No. 32 cooling lubricating oil, whose cooling channel is spiral shaped. The motor drives the spindle through a synchronous belt. The spindle is equipped with 7014 C angular contact ball bearings with DT installation method. The internal structure of the G1160 CNC machine tool spindle system is shown in Figure 1.

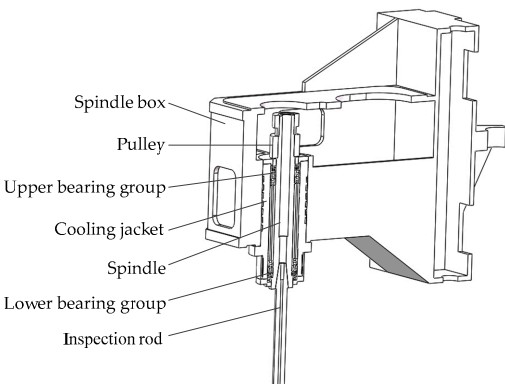

**Figure 1.** Internal structure of the spindle system.

The spindle runs at a speed of 6000 r/min. The intelligent thermal characteristic detection and compensation instrument for the CNC machine tool spindles is adopted to detect data of the spindle system. The working temperature of this instrument is 4–50 °C, and the sampling frequency is 5 s per time. The system temperature and ambient

temperature are detected through a PT100 temperature sensor manufactured by Heraeus in Germany. It has a 16 bit A/D converter and a measurement accuracy of 0.4%. The cooling oil temperature $T_{oil}$ of the CNC machine tool is detected and read using the RCO-15PTS oil cooler digital temperature display instrument which is equipped with the CNC machine tool and produced by Ruike Refrigeration Plant Co., Ltd, Dongguan, China. To detect the spindle deformation, a BT40 inspection rod with a diameter of 40 mm and a length of 300 mm is clamped onto the spindle. The axial deformation of the inspection rod is detected by the intelligent thermal characteristic detection and compensation instrument for CNC machine tool spindles, using the CPL230 sensor produced by LION PRECISION in America (Minneapolis, MN, USA). It includes a high-precision capacitive displacement probe and a compact multi-channel driver. As a non-contact sensor, its detection range is 125–375 µm and the working temperature is 4–50 °C. The RMS resolution of the sensor is 18 nm. The construction of an experimental platform for the data detection of the G1160 CNC machine tool spindle system and the instruments are shown in Figure 2.

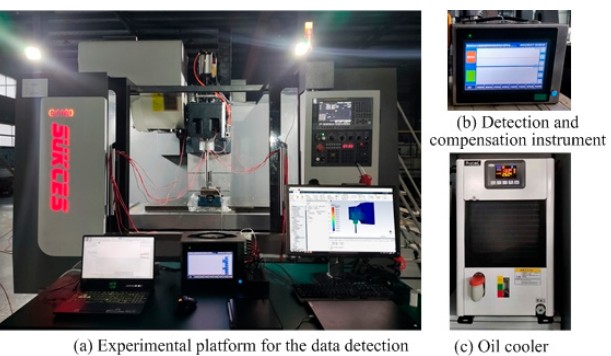

(b) Detection and compensation instrument

(a) Experimental platform for the data detection

(c) Oil cooler

**Figure 2.** Experimental platform and instrumentations.

Taking into account the structure of the spindle system, as well as the heat source and heat dissipation, the temperature sensors numbered T1–T8 were arranged according to Figure 3. The temperature values detected by the T1–T8 sensor are $T_i$, $i$ = 1, 2, ... The temperature data of the spindle end, spindle bearings, flange, spindle body, both sides of the spindle box, and spindle box support plate are detected separately. A temperature sensor numbered T9 is placed in the air to detect the ambient temperature $T_{air}$ simultaneously. The sensor used to detect the axial displacement $Z_s$ of the inspection rod is numbered Z. The settings for sensor positions are shown in Figure 3 and the description of the sensor positions is provided in Table 1.

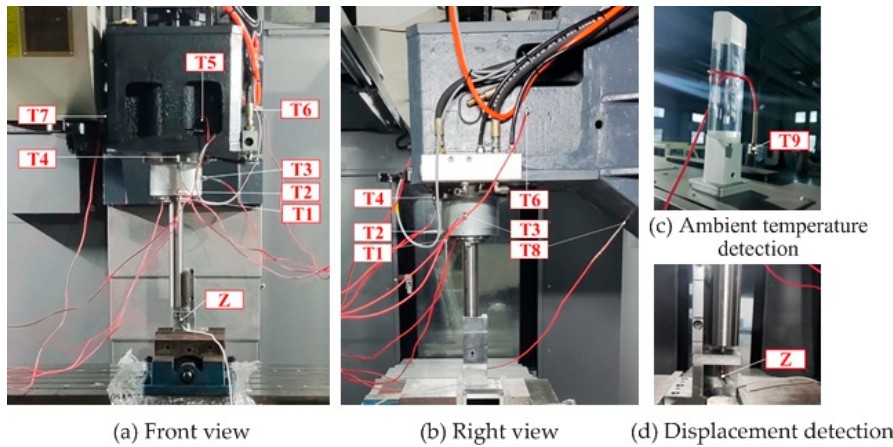

(a) Front view

(b) Right view

(c) Ambient temperature detection

(d) Displacement detection

**Figure 3.** Settings for sensor positions.

**Table 1.** Description of sensor positions.

| Sensor Type | Sensor Position | Sensor Number |
|---|---|---|
| | spindle end | T1, T2 |
| | spindle bearings | T3 |
| | flange | T4 |
| Temperature sensor | spindle body | T5 |
| | both sides of the spindle box | T6, T7 |
| | spindle box support plate | T8 |
| | ambient temperature | T9 |
| Displacement sensor | inspection rod end | Z |

The detected sample values of $T_{\text{air}}$ and $T_{\text{oil}}$ used for fitting at each time point are shown in Figure 4.

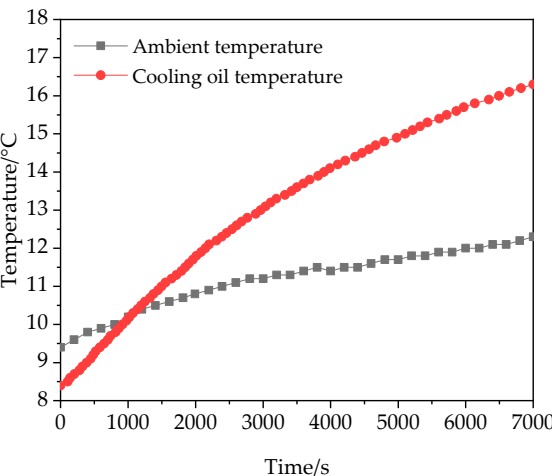

**Figure 4.** Detected ambient temperature and cooling oil temperature curves.

## 5. The Solution of Thermal Characteristics Based on Multiple Time-Varying Factors

It can be seen from Equations (10) and (11) that the friction torque generated by the lubricant is affected by the characteristics of kinematic viscosity. According to reference [26], the nonlinear time-varying characteristics of the No. 32 oil can be described using Equation (18).

$$v_{\text{oil}}(t) = 32 \exp[-0.0242 \times (T_{\text{oil}}(t) - 40)] \tag{18}$$

The torque $M_1$ generated by the load on the bearing can be calculated according to Equation (19) [27].

$$M_1 = f_1 P_1 d_{\text{m}} \tag{19}$$

where $f_1$ is a coefficient determined by the structure and load of the bearing, and $P_1$ is the calculated load for the bearing torque. For angular contact ball bearings, $f_1$ and $P_1$ can be calculated according to Equation (20) [28].

$$\begin{cases} f_1 = 0.0013(F_0/C_0)^{0.33} \\ P_1 = F_{\text{a}} - 0.1F_{\text{r}} \end{cases} \tag{20}$$

where $F_0$ and $C_0$ are the equivalent static load and rated static load of the bearing, and $F_{\text{r}}$ and $F_{\text{a}}$ are the radial and axial load of the bearing. $F_0$ is calculated according to Equation (21).

$$F_0 = X_0 F_{\text{r}} + Y_0 F_{\text{a}} \tag{21}$$

where $X_0$ and $Y_0$ are equivalent static load coefficients.

If $P_1$ obtained from Equation (20) is less than $F_{\text{r}}$, then $P_1 = F_{\text{r}}$. The load on the bearing of the spindle installing the inspection rod mainly includes the bearing preload

$F_p$, the synchronous belt compression spindle force $F_Q$, and the gravity $G$ of the rotating component. $F_p$ provided by the enterprise is 700 N. $G$ can be obtained by summing up the masses of each component. $F_Q$ is calculated according to Equation (22) [29].

$$F_Q = \frac{6K_A P_m}{\pi d_1 n_1} \times 10^7 \tag{22}$$

where $K_A$ is the operating condition coefficient, $P_m$ is the nominal power, $d_1$ is the diameter of the small pulley, and $n_1$ is the small pulley speed.

According to Figure 4, the ambient temperature rise range is only 2.9 °C during the detection period. The air property parameters in the convective heat transfer coefficient are all monotonic functions with air temperature as the independent variable. Equation (23) is constructed to determine the sensitivity of multiple air property parameters to the convective heat transfer coefficient. Low-sensitivity property parameters can be set as constants to simplify calculations.

$$S_g = \frac{|\Delta h|}{\overline{h}} = \frac{|h[g(T_{max})] - h[g(T_{min})]|t_e}{\int_0^{t_e} h(t)dt} \tag{23}$$

where $S_g$ is the sensitivity, $\Delta h$ is the extreme difference of the convective heat transfer coefficient affected by a certain property parameter within the detection temperature range, $\overline{h}$ is the time averaged value of the convective heat transfer coefficient during the experimental period, $g(T_{max})$ and $g(T_{min})$ are the peak values of a certain property parameter at the maximum and minimum detection values of experimental temperature, respectively, and $t_e$ is the end time of the experimental period.

According to reference [30], the natural convection heat transfer coefficient is much smaller than the forced convection heat transfer coefficient. Considering the actual radiation heat dissipation, the equivalent natural convection heat transfer coefficient is set to 9.7 W/(m²·°C).

$T_{air}(t)$ and $T_{oil}(t)$ are the important nonlinear time-varying factors affecting the thermal characteristics of the spindle system. The fitting degree of their mathematical models for experimental data can affect the accuracy of the calculation results. The fitting degree is evaluated using root mean square error (RMSE) and R-square. For this study, RMSE was calculated according to Equation (24) [31].

$$RMSE = \sqrt{\frac{1}{n}\sum_{i=1}^{n}\left[T(t_i) - \widetilde{T}(t_i)\right]^2} \tag{24}$$

where $n$ is the number of data samples, and $T(t_i)$ and $\widetilde{T}(t_i)$ are the detected value of the experiment and the fitting value of the mathematical model at the $i$-th time point.

The calculation of R-square is shown as Equation (25) [32].

$$R - square = \frac{\sum\limits_{i=1}^{n}\left[\widetilde{T}(t_i) - \overline{T}(t_i)\right]^2}{\sum\limits_{i=1}^{n}\left[T(t_i) - \overline{T}(t_i)\right]^2} \tag{25}$$

where $\overline{T}(t_i)$ is the average value of the detected temperature.

The rational function fitting method is used to fit the detected data of $T_{air}$ and $T_{oil}$. Both the Numerator degree and Denominator degree are set to 1. The fitted mathematical models are shown in Equation (26) and Equation (27), respectively.

$$\widetilde{T}_{air}(t) = \frac{14.02t + 44990}{t + 4777} \tag{26}$$

$$\widetilde{T}_{\text{oil}}(t) = \frac{24.99t + 60980}{t + 7435} \tag{27}$$

where $\widetilde{T}_{\text{air}}(t)$ and $\widetilde{T}_{\text{oil}}(t)$ are the fitted mathematical models of ambient temperature and cooling oil temperature. The *RMSE* and *R-square* of $\widetilde{T}_{\text{air}}(t)$ and $\widetilde{T}_{\text{oil}}(t)$ are shown in Table 2, which reflects a high degree of fit in $\widetilde{T}_{\text{air}}(t)$ and $\widetilde{T}_{\text{oil}}(t)$.

**Table 2.** *RMSE* and *R-square* of $\widetilde{T}_{\text{air}}(t)$ and $\widetilde{T}_{\text{oil}}(t)$.

| Function | *RMSE* | *R-Square* |
|:---:|:---:|:---:|
| $\widetilde{T}_{\text{air}}(t)$ | 0.9948 | 0.05874 |
| $\widetilde{T}_{\text{oil}}(t)$ | 0.9997 | 0.04025 |

According to the numerical table of thermophysical properties of air in Appendix 2 of Reference [33], the values of each parameter at different temperatures can be obtained by linear interpolation. According to Equation (23), the sensitivity solution results of air property parameters are shown in Table 3. The small variation in air property parameters, especially $\text{Pr}_{\text{air}}$, due to the small range of ambient temperature rise detected in this study can be seen from the table.

**Table 3.** The sensitivity of property parameters.

| Parameter | $\rho_{\text{air}}$ | $\mu_{\text{air}}$ | $\text{Pr}_{\text{air}}$ | $\lambda_{\text{air}}$ |
|:---:|:---:|:---:|:---:|:---:|
| Sensitivity/% | 0.69 | 0.49 | 0.01 | 0.87 |

The SST k-Omega turbulence model was used for solving in this study. The solution process adopts the SIMPLE strategy. In transient calculations, the values of time-varying factors and their parameters at each time step will be solved and updated by the solver in two ways: the time-varying factors are directly solved by substituting time steps into their time functions or the time function of property parameters; the factor values for the next time step are solved and updated based on the temperature values obtained from the current time step. It is worth noting that the different materials of different components and the different thermal characteristics of the same component at different positions can affect the waste heat dissipation after the last operation of the CNC machine tool. Therefore, there is a certain gradient in the initial temperature field of the spindle system, for which the initial detected temperature varies at different detection positions. Due to the focus of this study on the process of multiple factors affecting thermal characteristics, before solving, the spindle system is initialized with 7.4 °C, which is the lowest initial temperature value of each sensor detection position.

## 6. Analysis of Solution Results

### 6.1. Temperature Analysis of Spindle System

The temperature distribution on the surface and inside of the spindle system under the comprehensive effect of multiple nonlinear time-varying factors at 7000 s is shown in Figure 5. It can be seen that the high-temperature area is mainly distributed at the positions of the upper and lower bearings. Due to their influence, the temperatures of the spindle body, pulley, cooling jacket, and the root of the inspection rod are also relatively high. The temperature peak of the entire system is 30.8 °C, which is mainly near the installation position of the bearings. The high temperature value extends from this position and gradually decreases. The spindle body maintains around 20 °C because of its position in the middle of the upper and lower bearings. Due to the installation surface near the lower bearing group, the temperature at the root of the inspection rod is about 22 °C or above. The temperature of partial areas on the front, left, and right sides of the box is increased through the thermal conductivity of the connecting rib plates.

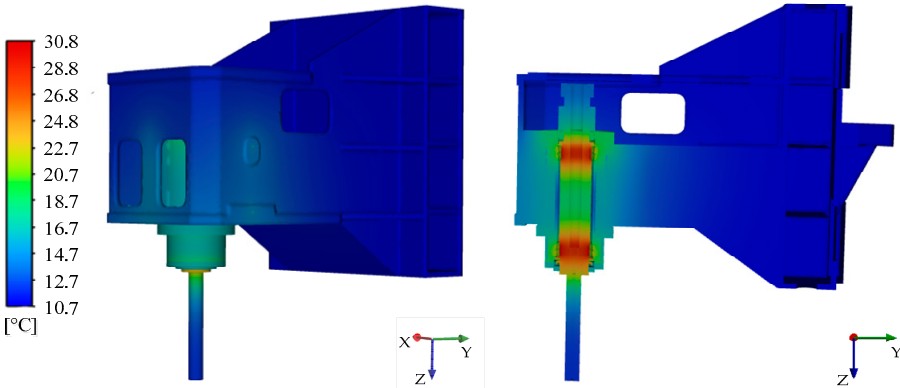

**Figure 5.** Temperature distribution on the surface and inside of the spindle system.

The temperature of the points numbered $\widetilde{T}1 \sim \widetilde{T}8$, which correspond one-to-one with the positions of sensors numbered T1~T8 on the spindle system, is solved at each time point. The temperature values are $\widetilde{T}_i$, $i = 1, 2, \ldots$ The curves of the experimental detection value and the solution value at each point are plotted. The relative error $\delta_i$ is calculated according to Equation (28), which is drawn to evaluate the accuracy of the model for solving the nonlinear time-varying thermal characteristics of the spindle system considering the comprehensive effects of internal and external factors.

$$\delta_i = \frac{T_i - \widetilde{T}_i}{T_i} \tag{28}$$

From the comparison shown in Figure 6, it can be seen that the experimental detection values and the solution values gradually increase over time. Their overall trend remains consistent. The relative error gradually decreases at each moment. Except for the several initial moments with relatively large relative errors, as time progresses, the relative error values of most points are within ±5% after 3000 s. At the final moment, the relative error of each point is below ±3%, and more than 60% of the points even reach within ±1%.

By analyzing the temperature curves of each point, there may be four reasons for the existence of relative error. Firstly, there are significant differences in the initial temperature values of each detection point which, to some extent, affect the temperature changes in the initial stage. Also, due to the small initial temperature values, the relative error is relatively large. However, it can be seen that the difference in initial temperature has a relatively short impact time. The overall fit between the solution value and the detection value of the temperature is relatively high. Secondly, $T_{\text{air}}$ detected by the sensor T9 for detecting ambient temperature has difficulty describing the actual complex temperature field around the spindle system. Due to the interference and gradient at different positions, the temperature inside and outside the system is different, which results in a difference in convective heat transfer coefficients. Thirdly, the oil temperature detected by the oil cooler digital temperature display instrument experiences a temperature drop during the entire pipeline transportation process due to unstable heat dissipation at different moments, which to some extent affects the solution results. Finally, there are complex variations in the characteristics of sealed components and contact thermal resistance between multiple components, as well as other minor time-varying factors that have not been considered. These changes affect the system temperature to varying degrees as the system temperature rises. Especially during the stage of rapid temperature rise, the actual temperature values are higher than the solution values and the relative error curve exhibits certain fluctuations. However, overall, the model for solving the nonlinear time-varying thermal characteristics of the spindle system has high accuracy.

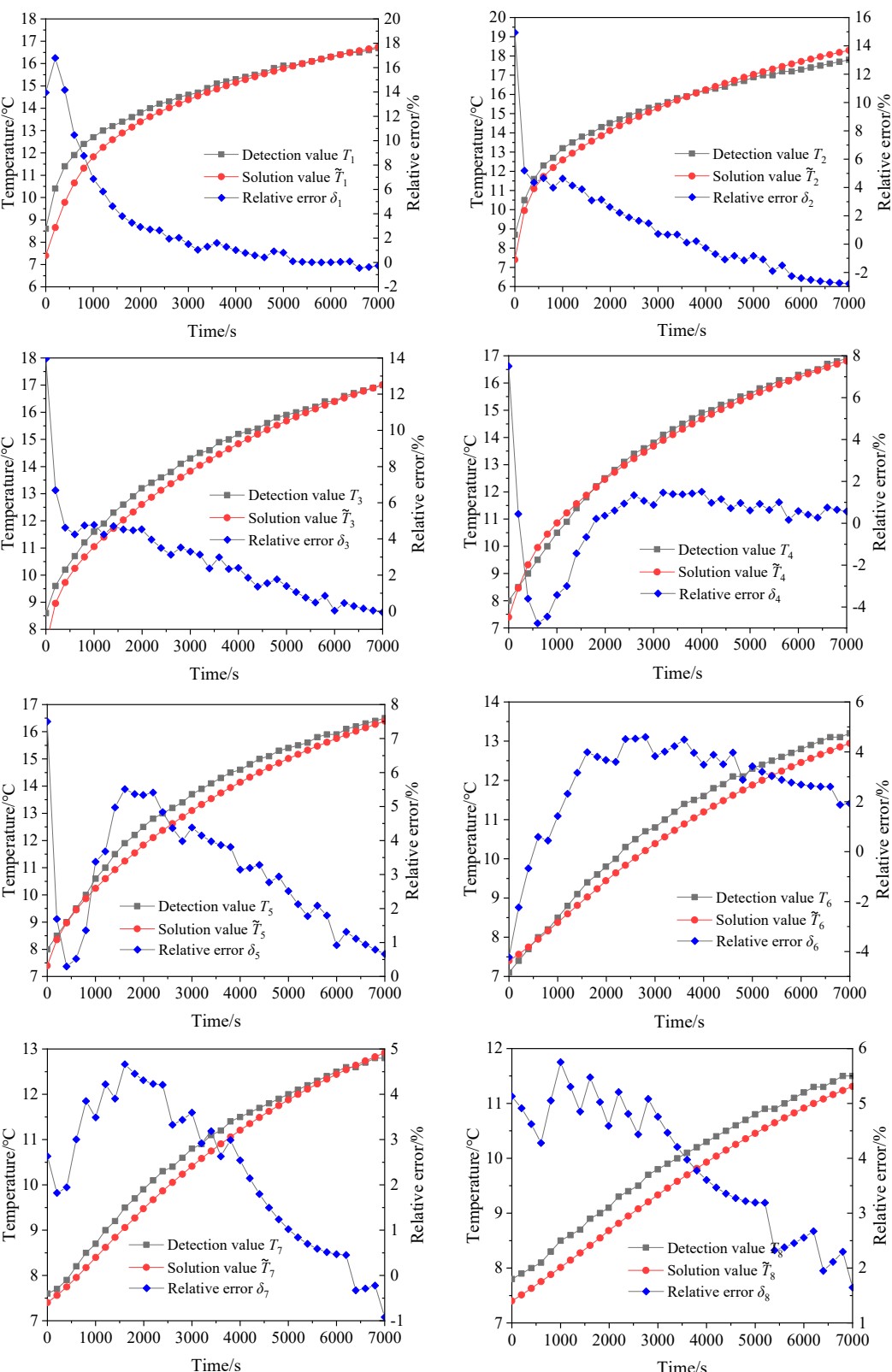

**Figure 6.** Temperature and relative error curves of each point.

### 6.2. Axial Displacement of the Spindle

The temperature field solution data at different time points are loaded onto the spindle system structure as thermal load conditions. In addition, the bearing preload and BT40 spindle tension, with the values provided by the enterprise, are loaded as the mechanical

conditions. The transient solution for the deformation of the spindle system has been completed. And the axial displacement $\widetilde{Z}_s$ of the inspection rod is simultaneously calculated. The total deformation of the spindle system at the final moment is shown in Figure 7. The relative error $\delta_s$ is calculated according to Equation (29). The comparison between the solution value $\widetilde{Z}_s$ and the experimental detection value $Z_s$ of axial displacement at the end of the inspection rod, as well as the relative error $\delta_s$ curve between the two, are shown in Figure 8.

$$\delta_s = \frac{Z_s - \widetilde{Z}_s}{Z_s} \tag{29}$$

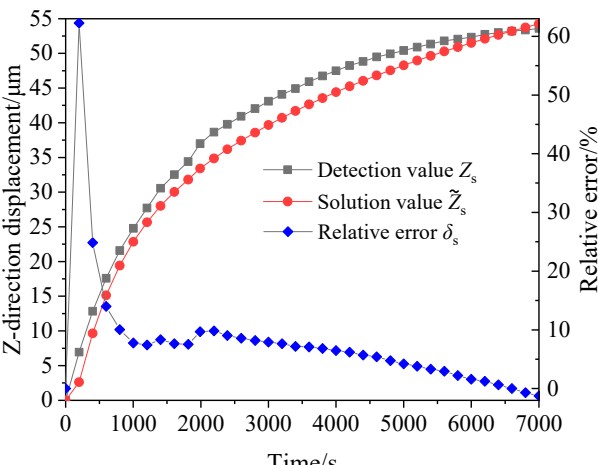

**Figure 7.** The total deformation of the spindle system.

**Figure 8.** The axial displacement and the relative error curve.

The installation surfaces of the spindle system are located on the back of the box. At the final moment, the total deformation of the spindle system shown in Figure 7 continuously accumulates as the distance from the installation surface increases. The maximum deformation occurs at the end of the inspection rod, which is 77.6 μm.

It can be seen from Figure 8 that both the detection value and the solution value of axial displacement at the end of the inspection rod gradually increase, and the trend of change is almost consistent throughout the entire solution period. Except for the initial few time points where the relative errors are large because of the small displacement values, the relative error of displacement after about 10 min is all within 10% and gradually approaching −1%. Due to the relative error in solving the temperature of the spindle system, the solution value of axial displacement at the end of the inspection rod is also smaller than the actual displacement. But over time, the values of the two gradually approach. This also fully reflects that the establishment of the thermal characteristics

solving model and the construction of multiple nonlinear time-varying factors for the spindle system has high rationality and accuracy.

## 7. Comparison of Thermal Characteristics Solution with Non Time-Varying Factors

The traditional methods often consider changes in the heat source conditions of the spindle system but rarely comprehensively consider the external environment, internal cooling conditions, and the heat transfer ability caused by their temperature during the actual operation of the CNC machine tool. Therefore, the ambient temperature and cooling oil temperature of the spindle system are set as non time-varying values in this study. In order to compare the thermal characteristics value at the last moment, separately, the ambient temperature and cooling oil temperature are set to the detected 12.3 °C and 16.4 °C, respectively. The thermal characteristics of each point in the spindle system are solved. The temperature comparison of the detection values and solution values affected by time-varying factors and by non time-varying factors at the final moment, as well as their relative errors, are shown in Figure 9.

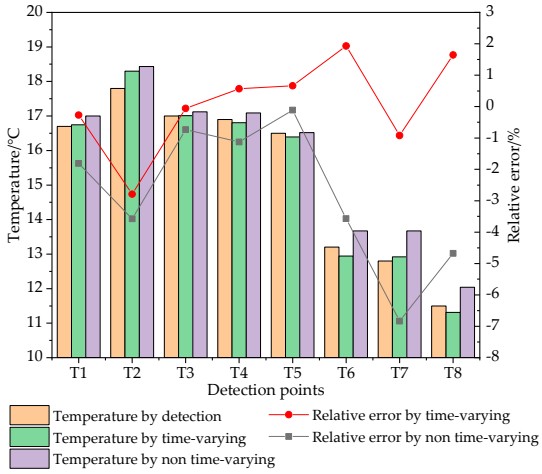

**Figure 9.** Temperature comparison of 8 points at the final moment.

It can be seen from Figure 9 that except for T5, the relative errors of the solution values affected by time-varying factors are closer to 0 than the solution values affected by non time-varying factors. This indicates that the proposed solution method considering the comprehensive effect of time-varying factors has more advantages in solving the accuracy of temperature in the spindle system.

The axial displacement curves of the detection values and solution values affected by time-varying factors and by non time-varying factors, as well as their relative error curves are shown in Figure 10. It can be seen that due to the constant ambient and cooling conditions, the axial displacement curve affected by non time-varying factors takes a relatively short time to approach a steady state, about 2500 s. However, in reality, the steady state is not achieved even at 7000 s. The change trend of axial displacement is not as close to the actual axial displacement rise, as solved by considering the effect of time-varying factors. From the perspective of calculation accuracy, at 7000 s, the relative error of the solution value calculated by non time-varying factors is −5.13% The value obtained by time-varying factors is −1.24%. The above analysis indicates that the nonlinear time-varying thermal characteristics model has advantages in reflecting the trend of numerical changes and the accuracy of result solving.

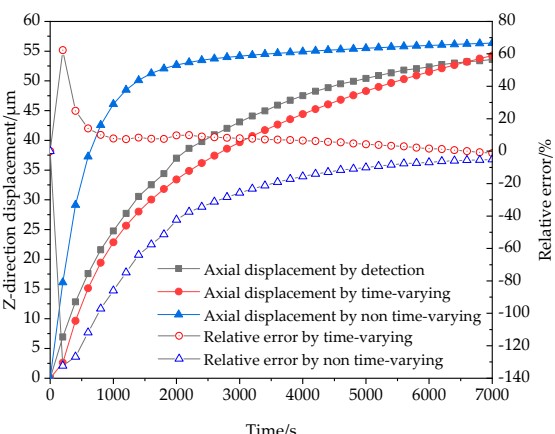

**Figure 10.** Comparison of axial displacement and relative error curves.

## 8. Conclusions

In this paper, a model for solving the nonlinear time-varying thermal characteristics of the spindle system is established based on numerical solution methods. And the nonlinear time-varying models of internal and external factors are constructed through theoretical derivation and data fitting. The temperature and deformation of the spindle system are solved taking into account the comprehensive effect of multiple factors. By comparing solution results affected by multiple nonlinear time-varying factors and by non time-varying factors with experimental data, the following conclusion can be drawn: the nonlinear time-varying thermal characteristics model has advantages in the solution of temperature and deformation for the spindle system over the method considering non time-varying factors; the established model for solving the nonlinear time-varying thermal characteristics of the spindle system has high rationality in solving the temperature field values of the spindle system; the construction method of mathematical models for nonlinear time-varying factors including friction torque generated by the lubricant, convective heat transfer coefficient, and coolant and ambient temperature is correct; and the solution results for the temperature and axial displacement of the spindle system by substituting multiple nonlinear time-varying factors into the nonlinear time-varying thermal characteristics model are almost consistent with the actual values. The relative errors of temperature and axial displacement are within ±3% and close to –1%, respectively. The thermal characteristics study of the spindle system based on the comprehensive effect of multiple nonlinear time-varying factors provides a theoretical basis for solving the thermal characteristics of spindle systems under complex working conditions.

**Author Contributions:** X.L. and X.D. carried out experiments and wrote the manuscript; J.Z. completed software operations and simulation calculations; X.Y. and H.S. checked the manuscript and revised it. All authors have read and agreed to the published version of the manuscript.

**Funding:** This research was financially supported by the National Natural Science Foundation of China (52175472; 62302263), Zhejiang Provincial Natural Science Foundation of China (LGG22E050031; LD24E050011; ZCLTGS24E0601), Natural Science Foundation of Zhejiang Province for Distinguished Young Scholars (LR22E050002), Science and Technology Plan Project of Quzhou (2022K90; 2021K41), Guiding Technology Research Project of Quzhou (ZD2022187; ZD2022188; ZD2022189).

**Data Availability Statement:** The data that support the findings of this study are available from the corresponding author upon reasonable request.

**Conflicts of Interest:** The authors declare no potential conflicts of interest with respect to the research, authorship, and/or publication of this article.

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
