# Peer review of "Thermal Characteristics of Spindle System Based on the Comprehensive Effect of Multiple Nonlinear Time-Varying Factors"

_processes, doi:10.3390/pr12020423_

Round 1
Reviewer 1 Report
Comments and Suggestions for Authors
Line 78: The need for research is derived from the fact that there are few reports on multiple nonlinear time-varying factors. If the reports are good, there is no need for additional research. Please make sure that the deficiencies in the state of the art are clear. What are the existing reports missing?
Line 87 - Please state here, what kind of discretization method (element type), solver (numerical method) and software will be used to solve all equations?
Line 142 - What method is used to assemble the model? Can you provide a picture of the inside of the spindel?
Line 231 - Who is the manufacturer of the G1160?
Line 234 - What is the ABB installation method?
Line 243 - What is "the intelligent thermal characteristic detection and compensation instrument for CNC machine tool spindles"?
Line 244 - Is it a capacitive sensor? Is it used for axial displacemen Z in Figure 2?
Table 2 and 3 - Instead of two very long tables, insert one diagramm with two curves
Line 328 - how did you combine the CFD model and the structural model?
Line 332 - Why can you not start with a cool system that did not run over night?
Figure 3 - can you provide a section view of the spindle? Is this thermal steady state? What is the spindle speed? Is it air cut?
Line 341 - How many parameters does your model have? What are they and where did you get them from? How uncertain are they? Did you do a sensitivity analysis regarding their influence on the model behaviour?
Figure 4 - I know it is in the text, but please make clear what is simulated and measured in the diagram.
Figure 5 - Please add coordinate system. Also reduce the digits of the scale.
Line 421 - Please add a comparison to the state of the art! What is better now? Can you provide a comparison with conventional modelling technics?
References: The international references need to be extended. Significant work on thermal issues has been done by the Prof. Christian Brecher (Germany), prof. Steffen Ihlenfeldt (Germany), prof. Konrad Wegener (Switzerland), prof. Jörg Luderich (Germany), Martin Marez (Czech Republic) - check out the following conference proceedings: https://link.springer.com/book/10.1007/978-3-031-34486-2
Reviewer 2 Report
Comments and Suggestions for Authors
In this paper, a model for solving nonlinear time-varying thermal characteristics in spindle systems is established. This model incorporates various factors including friction torque generated by lubricant, convective heat transfer coefficient, coolant, and ambient temperature. The model's potential application in predicting spindle deformation in other machine tools is highlighted.
Concerns and questions regarding this paper are raised as follows:
1-As most equations in this paper are cited from published works, the specific contributions of the current study need clarification.
2-Details regarding the intelligent thermal characteristic detection and compensation instrument are requested for a clearer understanding of its implementation.
3-Confusion arises from Table 2, particularly regarding the significant change in ambient temperature. Standard measurements in machine tools, as per ISO standards, typically utilize a temperature setting of around 20 degrees Celsius, whereas this work reports a maximum temperature of only 12.3 degrees Celsius. Clarification on this discrepancy is sought.
4-The importance of thermal balance in validating the proposed model is noted. Given the testing setup, with a maximum measurement time of approximately 7000 seconds, an increasing trend in thermal and displacement data is observed. However, the results indicate that thermal balance was not achieved, raising concerns about the model's performance.
5-For Equations 25 and 26, discrepancies are observed when t=0, as the results do not align with the values in Table 4. An explanation for this inconsistency is requested.
6-Attention is drawn to the positioning of Figure 3, which should be placed below line 350 for appropriate context.
7-Further details on the methodology used to generate the results presented in Figure 3 are requested.
8-Clarification is sought on the use of the CPL230 displacement sensor, produced by LION PRECISION, in obtaining the results shown in Figure 5.
9-Pay attention to the structure of this paper.
Comments on the Quality of English LanguageThe paper is largely clear and concise, effectively communicating the research. Some sentences, however, could be made more concise. Eliminating redundant words or phrases will enhance the paper's clarity and readability.
Round 2
Reviewer 1 Report
Comments and Suggestions for Authors
The authors response is comprehensive with valuable information. Thank you very much for that. I would like to see response 10, 12, 15 represented in the paper. I did not ask the question just for my own pleasure but also to be clarified in the paper.
Regarding comment 12 - I would like to see a list of time varying parameters and their values over time. That would provide some valuable insight for the reader. Also please explain how you would update the parameters in a practical context.
Figure 4 and 5 could be merged into one.
Reviewer 2 Report
Comments and Suggestions for Authors
The authors have solved my concerns. Therefore, I recommended to accept this paper for publication.
